# Radiosensitizing Effect of Trabectedin on Human Soft Tissue Sarcoma Cells

**DOI:** 10.3390/ijms232214305

**Published:** 2022-11-18

**Authors:** Mauro Loi, Giulia Salvatore, Michele Aquilano, Daniela Greto, Cinzia Talamonti, Viola Salvestrini, Maria Elena Melica, Marianna Valzano, Giulio Francolini, Mariangela Sottili, Costanza Santini, Carlotta Becherini, Domenico Andrea Campanacci, Monica Mangoni, Lorenzo Livi

**Affiliations:** 1Radiation Oncology Unit, Azienda Ospedaliero Universitaria Careggi, 50134 Florence, Italy; 2Department of Experimental and Clinical Biomedical Sciences “Mario Serio”, University of Florence, 50121 Florence, Italy; 3CyberKnife Center, Istituto Fiorentino di Cura e Assistenza (IFCA), 50139 Florence, Italy; 4Department of Orthopaedic Oncology, Azienda Ospedaliero Universitaria Careggi, 50134 Florence, Italy

**Keywords:** soft tissue sarcoma, trabectedin, radiation, radiosensitizers

## Abstract

Trabectedin is used for the treatment of advanced soft tissue sarcomas (STSs). In this study, we evaluated if trabectedin could enhance the efficacy of irradiation (IR) by increasing the intrinsic cell radiosensitivity and modulating tumor micro-environment in fibrosarcoma (HS 93.T), leiomyosarcoma (HS5.T), liposarcoma (SW872), and rhabdomyosarcoma (RD) cell lines. A significant reduction in cell surviving fraction (SF) following trabectedin + IR compared to IR alone was observed in liposarcoma and leiomyosarcoma (enhancement ratio at 50%, ER50: 1.45 and 2.35, respectively), whereas an additive effect was shown in rhabdomyosarcoma and fibrosarcoma. Invasive cells’ fraction significantly decreased following trabectedin ± IR compared to IR alone. Differences in cell cycle distribution were observed in leiomyosarcoma and rhabdomyosarcoma treated with trabectedin + IR. In all STS lines, trabectedin + IR resulted in a significantly higher number of γ-H2AX (histone H2AX) foci 30 min compared to the control, trabectedin, or IR alone. Expression of ATM, RAD50, Ang-2, VEGF, and PD-L1 was not significantly altered following trabectedin + IR. In conclusion, trabectedin radiosensitizes STS cells by affecting SF (particularly in leiomyosarcoma and liposarcoma), invasiveness, cell cycle distribution, and γ-H2AX foci formation. Conversely, no synergistic effect was observed on DNA damage repair, neoangiogenesis, and immune system.

## 1. Introduction

Soft-tissue sarcomas (STSs) are a heterogeneous group of tumors that arise from mesenchymal cells, encompassing over 50 different histologic subtypes. Despite their rarity, STSs pose a significant health problem due to their aggressive behavior and poor prognosis, with 55% overall survival at 5 years and limited effective therapeutic options [1]. In patients affected by the advanced stage of the disease, the current standard of care is chemotherapy, which includes drugs such as dacarbazine, doxorubicin, epirubicin, ifosfamide, gemcitabine, and taxanes, with only few novel agents approved in the last 30 years [1]. Hence, the development of novel, effective treatment strategies represents a huge clinical challenge. Among the new generation of drugs available in this setting, trabectedin (Yondelis^®^) is approved for the treatment of adult patients with advanced STS who are unsuitable for or progressing after anthracycline-based chemotherapy [2]. This compound presents a multifaceted mechanism of action affecting several key processes in both tumor cells and the microenvironment. On a cellular scale, trabectedin directly interacts with DNA to modulate gene expression through transcription inhibition, resulting in a potent cytotoxic effect [3]. Moreover, trabectedin exerts a significant influence on the tumor microenvironment (TME), inducing a downregulation in the expression of pro-inflammatory and pro-angiogenetic cytokines such as IL-6, angiopoietin 2 (Ang-2), and vascular endothelial growth factor (VEGF) [4]. It has also been proposed that trabectedin may alter the phenotypic profile of immune-resident cells, selectively reducing both monocytes and tumor-associated macrophages (TAMs), a population of macrophages involved in tumor immune evasion [5], in treated tumors. Moreover, trabectedin activity on TAMs may cooperate with immune checkpoint inhibitors directed toward programmed death-ligand 1 (PD-L1) to re-establish immune surveillance according to recent reports [6]. In summary, trabectedin may combine direct cytotoxic activity toward cancer cells with the peculiar capacity to favorably modify the TME and potentially counterbalance tumor-related immunosuppression. Trabectedin has been shown to enhance the cytotoxic effect of IR in preclinical models of solid tumors [7,8]. In vitro studies suggested an intrinsic radiosensitizing activity of trabectedin possibly linked to effects on cell cycle progression. Interestingly, trabectedin mainly targets DNA through the constitution of adducts that mimic interstrand cross-links, triggering the formation of DNA strand breaks during the processing by the DNA repair machinery [9]. The cytotoxic activity of trabectedin may be influenced by DNA damage repair pathways such as nucleotide excision repair (NER) and homologous recombination (HR) involved in radiation-induced lesion recognition. Indeed, trabectedin exhibits reduced efficacy in cells deficient in NER, while it is more potent in homologous-recombination-deficient cells [10]. However, a possible synergistic effect of combined IR and trabectedin has not yet been demonstrated in vitro. Similarly, it is unknown whether trabectedin-induced TME modification may increase response to radiotherapy by abolishing TAMs-related radioresistance and modulating angiogenesis to a normoxic vascular architecture.

In this study, we hypothesize that trabectedin could enhance the efficacy of IR by increasing cell radiosensitivity and modulating the tumor vascular and immune micro-environment in STS in vitro models. In order to assess variability among different sarcoma subtypes, we performed our experiments on different STS human cell lines, representing four different subtypes.

## 2. Results

### 2.1. Trabectedin Is Cytotoxic in the Low Nanomolar Range in STS Cells

We found that the IC50 of trabectedin was 1.296 nM for LMS, 0.6836 nM for LPS, 0.9654 nM for RMS, and 0.8549 nM for FS. Then, cell viability is reported as a percentage versus the control (Figure 1).

### 2.2. Trabectedin Improves Cytotoxic Effect of IR

Among all the STS cell lines, LMS and LPS showed a significant reduction of SF after trabectedin + IR compared to IR alone, both at 4 Gy and 6 Gy (Figure 2). LMS showed the highest significant radiosensitization with an ER50 of 2.35, whereas LPS showed an ER50 of 1.45. Both FS and RMS did not show a significant difference in SF when treated with trabectedin + IR compared to control. The observed SF in the combined treatment of trabectedin + IR in LMS and LPS cells was significantly lower than the expected SF, suggesting a synergistic effect between trabectedin and IR on clonogenic cell death. Conversely, RMS and FS showed an additive effect (Figure 3).

### 2.3. Trabectedin Significantly Reduces Invasiveness Caused by IR

All STS cell lines showed a markedly reduced invasion capability, expressed as a significant reduction in invasive cell count, when treated with trabectedin alone or trabectedin + IR compared to the control. Moreover, a statistically significant reduction of invasive cells in the trabectedin + IR sample compared to IR alone was observed. Furthermore, all STS cell lines showed a higher number of invasive cells in IR alone compared to trabectedin alone, although the number of invasive cells was significantly lower than the control. Interestingly, in LMS, radiation did not significantly decrease the number of invasive cells compared to the control; yet, the addition of trabectedin + IR significantly reduced the number of invasive cells (Figure 4 and Appendix A).

### 2.4. Trabectedin Affects Cell Cycle Progression of STS Cells

Trabectedin alone showed a non-specific perturbation in the cell cycle progression profile in FS and RMS, whereas it seemed to have no effect in LPS and LMS compared to the control. In IR alone, we observed a slight increase (5%) of the cells in the G2/M phase in RMS and FS. Interestingly, we found a strong increase (+20%) of G2/M-phase cells in IR alone in LPS compared to the control, suggesting an enhanced cell cycle arrest in the G2/M phase. Instead, LMS treated with IR alone did not change the cell cycle profile compared to the control. Trabectedin + IR treatment showed a significant reduction in G2/M-phase cells in both LMS and RMS, whereas in LPS, it was associated with a modest increase, although inferior to IR alone. In FS, the combined treatment seemed not to modify cell cycle redistribution (Figure 5).

### 2.5. Trabectedin Combined with IR Significantly Increased the Number of γ-H2AX Foci

In all the STS cell lines, 30 min after irradiation, a significantly higher intensity of fluorescence emitted by γ-H2AX foci was observed following trabectedin + IR as compared to the control, trabectedin, and IR alone. In LPS and RMS treated with IR alone, the number of foci significantly increased 30 min after treatment and decreased over time. In all cell lines, the number of foci decreased in all the treatment groups over time. However, 24 h after treatment, the intensity of the fluorescence emitted by the γ-H2AX foci in the trabectedin + IR group was still significantly higher as compared to the control, suggesting a recovery control (Figure 6 and Appendix A).

### 2.6. Trabectedin Combined with IR Affects the Expression of DNA Damage Response Proteins, Angiogenic Factors, and Immune Checkpoint Proteins

In LMS, 30 min after IR, ATM and RAD50 expression significantly increased in IR alone and trabectedin + IR as compared to the control and trabectedin alone. After 1 h and 2 h of treatment, ATM significantly decreased in IR and trabectedin alone, as compared to control. At these two time points, RAD50 did not show significant changes. Conversely, 24 h after treatment, RAD50 expression significantly decreased in IR alone compared to the control, whereas no variations in ATM levels were observed. In LPS, ATM expression did not show significant changes at the different time points, whereas RAD50 significantly increased 30 min after treatment in trabectedin alone, radiation, and trabectedin + IR compared to the control. Therefore, radiation might induce DNA repair response immediately after 30 min of treatment and decrease over time. In LMS, VEGF expression significantly decreased in trabectedin and IR alone compared to control 2 h after treatment, whereas Ang-2 did not change. Twenty-four hours after treatment, both VEGF and Ang-2 were lower than the control, although VEGF expression significantly increased compared to the outcomes observed after 2 h of treatment. In the presence of trabectedin + IR, no modifications were observed in both VEGF and Ang-2 compared to the control. In LPS, VEGF did not undergo changes in expression, whereas Ang-2 significantly increased 24 h after treatment in IR alone and trabectedin + IR compared to control and significantly decreased in trabectedin + IR compared to IR alone. In LMS, a significant decrease in PDL1 was observed 24 h after treatment with trabectedin and IR alone compared to the control. In LPS, only trabectedin determined a decreased expression of PD-L1 compared to the control (Figure 7).

## 3. Discussion

In this study, we observed that trabectedin might enhance the efficacy of IR in preclinical models of STS. Several studies showed the effects of trabectedin as a single agent or in combination with other molecules [11,12]. The SF of eight human cell lines treated with trabectedin was significantly lower as compared to other clinically used chemotherapy agents, such as methotrexate, doxorubicin, etoposide, and paclitaxel, with IC50 values of trabectedin in picomolar ranges [13]. D’Incalci et al. reported synergistic activity of trabectedin with cisplatin [14], while Takahashi et al. [15] evaluated the combination of trabectedin with doxorubicin, trimetrexate, and paclitaxel in FS and LPS cell lines, reporting a cytotoxic activity of trabectedin alone in these cell lines and a synergistic effect with doxorubicin. These results have been translated in a recent phase II trial LMS-02 study (NCT02131480), which tested the combination of doxorubicin and trabectedin as the first-line systemic treatment for metastatic LMS, prompting a phase III trial comparing doxorubicin with or without trabectedin [16]. However, few data are available concerning the interaction between IR and trabectedin. Simoens et al. [17] reported a moderate increase in radiosensitivity in three (ECV304, H292, and CAL-27) out of four solid tumor cell lines treated with cytotoxic concentrations of trabectedin 24 h before radiation, correlated to a concentration-dependent G2/M blockade. Romero et al. [8] assessed the potential radiosensitizing effect of trabectedin on different solid tumor cell lines, determining a dose enhancement factor (DEF) of 1.92 and 1.77 at the IC50 dose level, respectively, in DU145 prostate cancer cells and HeLa ovarian cancer cells. A recent study reported increased γH2AX foci detection in in vitro models of lung and colon cancer [7]. Put together, these data might suggest that the synergistic activity of trabectedin and IR is found across different tumor types and may be partially explained by the synchronization of the cell cycle in a radiosensitive phase, resulting in a direct increase of DNA damage.

Focusing on STS, no published study to our knowledge specifically explored the interaction between IR and trabectedin in sarcoma cell lines, except for preliminary data presented at the 2017 Congress of the American Society for Radiation Oncology (ASTRO): in their poster, Jarboe et al. reported a cell-cycle-dependent radiosensitization in the leiomyosarcoma cell line SK-LMS-1, although this was not observed in the liposarcoma cell line SW872 [18]. Interestingly, clinical trials are already testing concurrent trabectedin-based chemoradiation in patients affected by STS [19,20]. Hence, advancement in translational research is urgently required to identify the most rational combination of drug and radiation dose and to identify disease subtypes that are expected to draw a higher benefit.

In this study, we investigated in vitro if trabectedin could enhance radiotherapy by modulating cell radiosensitivity and the biological behavior of different STS human cell lines, particularly in regard to cell cycle progression, invasiveness, DNA damage onset and repair capability, proangiogenic cytokine release, and immunosuppressive protein expression. Considering the heterogeneity in response to treatment across different histological subtypes, four STS subtypes were included in our study.

We found that the addition of trabectedin to radiation induced a significant reduction of SF in an in vitro model of different human STS subtypes, with a synergistic effect in LMS and LPS, respectively, at 4 Gy and 6 Gy, whereas RMS and FS showed an additive effect. The number of invasive cells was significantly reduced when treated with trabectedin alone or combined with radiation compared to the control, suggesting that a combined treatment of trabectedin + IR could counteract a possible influence of radiation in determining the onset of more aggressive tumor clones, as described by some authors [21].

Cell cycle analysis yielded different results according to the STS cell line. Trabectedin induced cell cycle perturbations in FS and RMS. The combination of trabectedin + IR affected cell cycle distribution only in LMS and RMS. These results suggest that trabectedin induces variable effects on the cell cycle depending on the STS subtype, suggesting that the radiosensitizing effects may not be completely explained by an increase in G2/M blockade.

In all four analyzed cell lines, 30 min after irradiation, the combination of trabectedin + IR resulted in a significantly higher γ-H2AX foci detection compared to control, trabectedin, and IR alone. After 24 h of treatment, γ-H2AX foci detection in the combined treatment decreased, but remained significantly higher compared to the control, thus suggesting a recovery mechanism.

ATM and RAD50 are both crucial upstream regulators of cellular cycle checkpoints and DNA repair response. ATM is an ionizing-radiation-activated protein kinase that signals DSBs in mammalian cells by phosphorylating and activating proteins involved in cell cycle checkpoint responses, such as p53 and BRCA1 [22]. RAD50 belongs to the MRN complex (Mre11/Rad50/Nbs1) required for cellular responses to DSBs and normal S-phase checkpoint function in mammalian cells. The DNA damage response is highly regulated, and the order in which ATM and MRN act in the early phase of the DSB response is still unclear. The MRN complex acts both upstream and downstream of ATM. The MRN complex rapidly identifies DSBs and assists in ATM activation. ATM, in turn, phosphorylates all the members of the MRN complex to initiate downstream signaling [23,24]. The association of ATM gene mutation/deletion with STS suggested a link of the ATM gene with cancer risk, and ATM kinase may contribute to the pathogenesis of STS [24,25]. Interestingly, Western blot analysis, in LMS and LPS, showed that trabectedin + IR did not decrease ATM and RAD50 expression compared to IR alone, suggesting that trabectedin might not affect radio-induced DNA damage repair response.

Since STS are highly vascularized, we measured vascular endothelial growth factor (VEGF) and angiopoietin 2 (Ang-2) to investigate if trabectedin combined with radiation affects tumor vascularization. VEGF induces angiogenesis and endothelial cell proliferation. Ang-2 antagonizes Ang1–Tie2 signaling, required for maintaining vessel integrity through the recruitment of peri-endothelial cells, and induces the loosening of vascular structure and an increase in the vascular permeability exposure of endothelial cells to pro-angiogenesis factor such as VEGF. High levels of VEGF-A in tumors and blood samples from STS patients are associated with a higher tumor grade, an increased tendency to form metastases, a reduced response to treatment, a lower overall survival, and an increased risk of recurrence [26].

An emerging therapeutic option in radiotherapy is immunotherapy. In sarcomas, several data suggested it is an interesting strategy, even if the knowledge of the immune response remains limited compared to other cancers. An unfavorable prognostic role of tumor-associated macrophages was reported in LMS and myxoid LPS. Immunotherapeutic strategies may be promising in sarcoma patients. Indeed, the heterogeneity of sarcomas makes the identification of a unified molecular pathway that could be exploited to treat a large proportion of them hard [27]. Finally, the capability of PD-L1 to exert a major role in suppressing antitumor adaptive immunity and pharmacological blockade using immune checkpoint inhibitors (ICI) is currently under investigation in STS [28]; interestingly, radiotherapy may induce the upregulation of its expression in certain tumors [29]. Trabectedin + IR did not alter either the expression of cytokines involved in neoangiogenesis nor the tumor immune evasion. However, an in vitro model may lack accuracy to reproduce the complex interplay among radiation, cancer cells, and TME in the living host and may require further investigation in in vivo models.

## 4. Materials and Methods

### 4.1. Reagents

Trabectedin was provided by PharmaMar, S.A. (Madrid, Spain). Reagents for cells cultures and Western blot and matrigel were purchased from Sigma-Aldrich (St. Louis, MO, USA). The BCA protein assay kit was obtained from Pierce Biotechnology (Rockford, IL, USA). Phospho-gamma-H2AX (pSer139) was purchased from Cell Signaling Technology (Danvers, MA, USA). Antibodies for ATM, RAD50, angiopoietin 2, VEGF, PDL1, STAT1, and GADPH were purchased from Santa Cruz Biotechnology, Inc. (Heidelberg, Germany). Peroxidase-conjugated secondary antibody was purchased from Calbiochem (Darmstadt, Germany). Anti-rabbit secondary antibody (Alexa Fluor 488) was purchased from Thermo Fisher Scientific (Waltham, MA, USA).

The ThinCert cell culture insert PET membrane for the cell invasion assay was obtained from Greiner bio-one (Kremsmünster, Austria). The DAPI Nuclear Stain was purchased from Merck Life Science S.r.l. (Darmstadt, Germany). The ibidi µ-Slide VI 0.4 was purchased from Ibidi Gmbh (Martinsried, Germany).

### 4.2. Cell Cultures

Human fibrosarcoma (FS; HS 93.T), leiomyosarcoma (LMS; HS5.T), liposarcoma (LPS; SW872), and rhabdomyosarcoma (RMS; RD) cell lines were obtained from ATCC (Manassas, VA, USA) and cultured in DMEM with 10% FBS, 2 mM L-glutamine, 100 U/mL penicillin, and 100 mg/mL streptomycin at 37 °C in a humidified 5% CO_2_ atmosphere.

### 4.3. Cytotoxic Assay

An MTS assay was used to measure IC50 (drug concentration resulting in 50% inhibition of cell growth). Values were determined by seeding 2000 cells/well in 100 µL of growth medium in 96-well plates and incubating cells overnight at 37 °C to allow attachment. After 24 h, the medium was changed without (control) or with increasing doses of trabectedin during 72 h. During incubation, the medium was not changed, nor was trabectedin added again. Then, 20 µL of reagent was directly added to the wells, and the plates were incubated at 37 °C for at least 1 h. Absorbance was measured at 490 nm on the VICTOR 3 1420 Multilabel Plate Reader (PerkinElmert, Inc., Waltham, MA, USA). Data from each treatment were reported as the percentage of the relative control.

### 4.4. Clonogenic Assays

For clonogenic assays, 1500, 2000, 2500, and 3000 cells/well were seeded in six-well plates. After 24 h, a single IR dose of 2 Gray (Gy) (cells plated at 2000), 4 Gy (cells plated at 2500), or 6 Gy (cells plated at 3000) was delivered, with or without trabectedin pretreatment. To assess differences in SF and avoid the confounding effect of potential cell overkill, trabectedin was administered at a dose corresponding to 12% of the IC50. Cells plated at 1500 cells/well were used as controls. Cells were processed as previously described [30]. Stained colonies were counted and normalized to the respective controls (trabectedin-treated cells for combined treatments). Linear-quadratic survival was fit to the data by nonlinear regression. The radiosensitization enhancement ratio for trabectedin at 50% survival (ER50) was calculated as follows: ER50 = dose at 50% survival without trabectedin/dose at 50% survival with trabectedin. Furthermore, according to the previously reported methodology [31], we compared the SF experimentally observed after combined treatment with the ‘‘expected’’ SF in the presence of a mere additive effect, in order to evaluate if the combination of trabectedin with radiation showed a synergistic effect.

### 4.5. Invasion Assay

Cell invasion assays were performed using the ThinCert cell culture insert PET membrane with an 8 µm pore size coated with 0.3 mg/mL of matrigel, a reconstituted basement membrane. Briefly, cells were trypsinized and counted, then cells were seeded in the upper chamber at 1 × 10^4^ cells/well in serum-free DMEM. DMEM supplemented with 10% FBS was placed in the bottom well as a source of chemoattractant. Incubation was carried out for 24 h at 37 °C in a humidified 5% CO_2_ atmosphere. At the end of the assay, non-invasive cells in the upper chamber were wiped off with a cotton swab, whereas cells on the lower surface were permeabilized by 100% methanol and stained with crystal violet. In each invasion assay, non-treated cells were taken as the controls. The number of invasive cells was counted in 8 visual fields to obtain an average value. Then, the data are reported also as the percentage of positive stained cells normalized on the control considered as 100%. Invasive cells were subsequently extracted with DMSO and read at 595 nm.

### 4.6. Flow Cytometry Analysis

Cells were synchronized by starvation with serum-free medium in the 24 h preceding irradiation. Then, cells were pre-treated with trabectedin and/or irradiated to a dose of 4 Gy. After 24 h, cells were harvested by trypsinization and re-suspended in 0.3% BSA PBS wash buffer. FS, RS, and LPS to a density of 5 × 10^5^ cells were fixed with 2% formaldehyde and then stained with DAPI Nuclear Stain. LMS cells were resuspended to a density of 1 × 10^5^ cells/mL in 0.3% BSA PBS wash buffer, then fixed in cold 70% ethanol for 1 h at 4 °C, and stained with 50 μg/mL of PI and 100 μg/mL of ribonuclease RNase, to ensure only DNA staining. Stained cells were analyzed using a MACSQuant Analyzer 10 benchtop flow cytometer (Miltenyi Biotec) according to the manufacturer’s instructions.

### 4.7. Immunofluorescence Analysis

Cells were treated as previously reported [31]. Briefly, cells were seeded on ibidi µ-Slide VI 0.4 in growth medium, then were starved with serum-free medium in the 24 h preceding irradiation. At this point, cells were pre-treated with trabectedin and/or irradiated at 4 Gy. The first samples were fixed 30 min after irradiation, and the remaining samples were incubated for 6 h and 24 h under standard cell culture conditions to allow repair of radiation-induced DNA double-strand breaks (DSBs). Then, cells were stained with the anti-gamma-H2AX antibody. A Leica TCS SP5-II laser confocal microscope (Leica, Milan, Italy) with a 40× objective was used to analyze the specimens. The mean immunofluorescent signal per nucleus emitted from gamma-H2AX was determined by the ImageJ software (NIH, Bethesda, MD, USA). The pan-nuclear signals without foci were excluded from the analysis. Then, the immunofluorescent signal resulting from each treatment is reported as the percentage of the immunofluorescent signal of the control cells.

### 4.8. Western Blot

Since the LMS and LPS cell lines were found to be more affected by trabectedin, we decided to perform further experiments on these two cell lines, to assess modifications in the expression of DNA damage response proteins (ATM, RAD50), proangiogenic cytokines (VEGF, Ang-2), and immune inhibitory signaling proteins (PD-L1).

Cells were lysed in 150 μL ice-cold RIPA buffer (150 mM NaCl, 50 mM TRIS pH 7.5, 1% (*v*/*v*) Nonidet P-40, 0.5% (*w*/*v*) sodium deoxycholate, 0.1% (*v*/*v*) SDS) supplemented with the protease/phosphatase inhibitor cocktail. Cells were centrifuged for 10 min at 9000 rpm at 4 °C, then the supernatants collected and the protein concentration measured. Protein aliquots (30 μg), processed and loaded onto 8 and 15% SDS-PAGE, were transferred on PVDF membranes, which were incubated overnight at 4 °C with primary antibodies diluted in Tween Tris-buffered saline, followed by peroxidase-conjugated secondary IgG. Proteins were revealed using the Immobilon Western Chemiluminescent HRP Substrate. The chemiluminescent signal was analyzed with the Image lab software on a ChemiDoc Touch Imaging system (Bio-Rad Laboratories). STAT1 and GADPH were used as the loading control. The densitometric analysis of Western blot bands was performed using the Quantity One software (Bio-Rad Laboratories, Inc, Milan, Italy), normalized to STAT1 and GADPH, and reported as the percentage relative to the respective control, set as 100.

### 4.9. Statistics

Statistical analysis was performed using one-way analysis of variance (ANOVA), and *p* ≤ 0.05 was considered statistically significant. Data are expressed as the means ± SEM.

## 5. Conclusions

These results suggest that trabectedin combined with radiation has an effect on cell survival, in particular in the LMS and LPS subtypes, on cell invasiveness, on progression through the different steps of the cell cycle depending on the cell type and dose, and on double-strand breaks’ formation. Conversely, in the LMS and LPS subtypes, no synergistic activity was demonstrated on DNA damage repair mechanisms, neo-angiogenesis, or immune evasion. Further investigations, in particular in in vivo models, are required to clarify the mechanisms underlying the potential synergistic effect of trabectedin combined with radiation in certain STS subtypes, in particular with regard to the molecular determinants of the intrinsic susceptibility of different histological subtypes, such as the specific interaction with disease-related translocations.

## Figures and Tables

**Figure 1 ijms-23-14305-f001:**
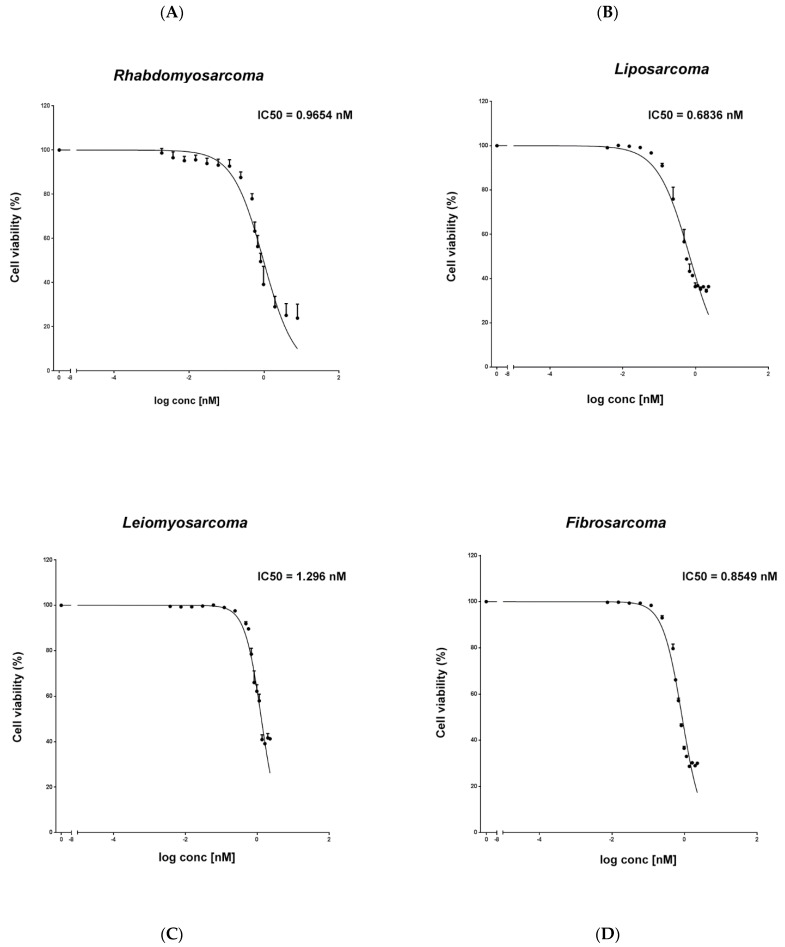
Evaluation of the cytotoxic effect of trabectedin in STS cell lines. An MTS assay was used to determine cell viability of rhabdomyosarcoma RD (**A**), liposarcoma SW872 (**B**), leiomyosarcoma HS5.T (**C**), and fibrosarcoma HS 93.T (**D**) cells treated with trabectedin.

**Figure 2 ijms-23-14305-f002:**
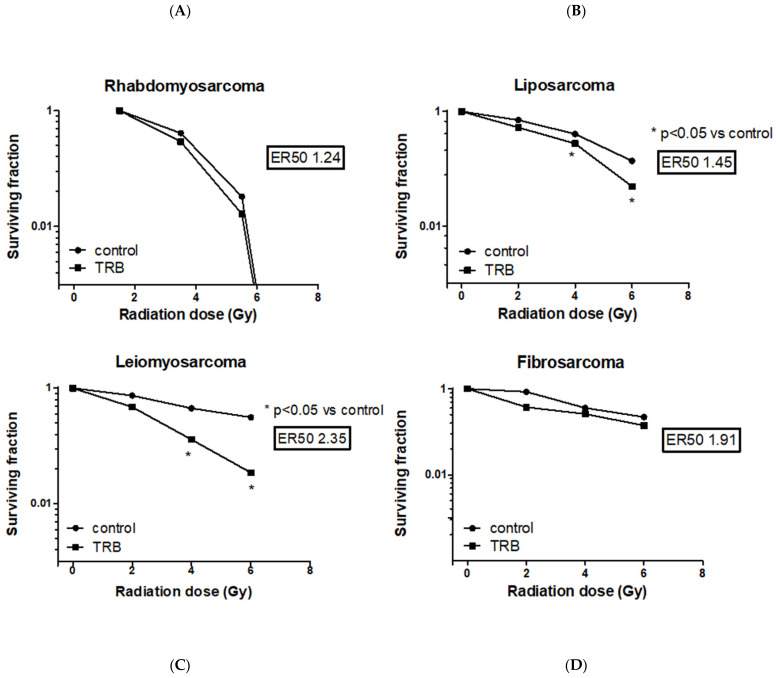
Evaluation of surviving fraction after irradiation with increasing doses in rhabdomyosarcoma RD (**A**), liposarcoma SW872 (**B**), leiomyosarcoma HS5.T (**C**), and fibrosarcoma HS 93.T (**D**) cell lines, respectively, the control, pretreated with trabectedin, were irradiated with 0–6 Gy. Surviving fraction is shown as the mean ± SEM. * *p* ≤ 0.05 vs. the control.

**Figure 3 ijms-23-14305-f003:**
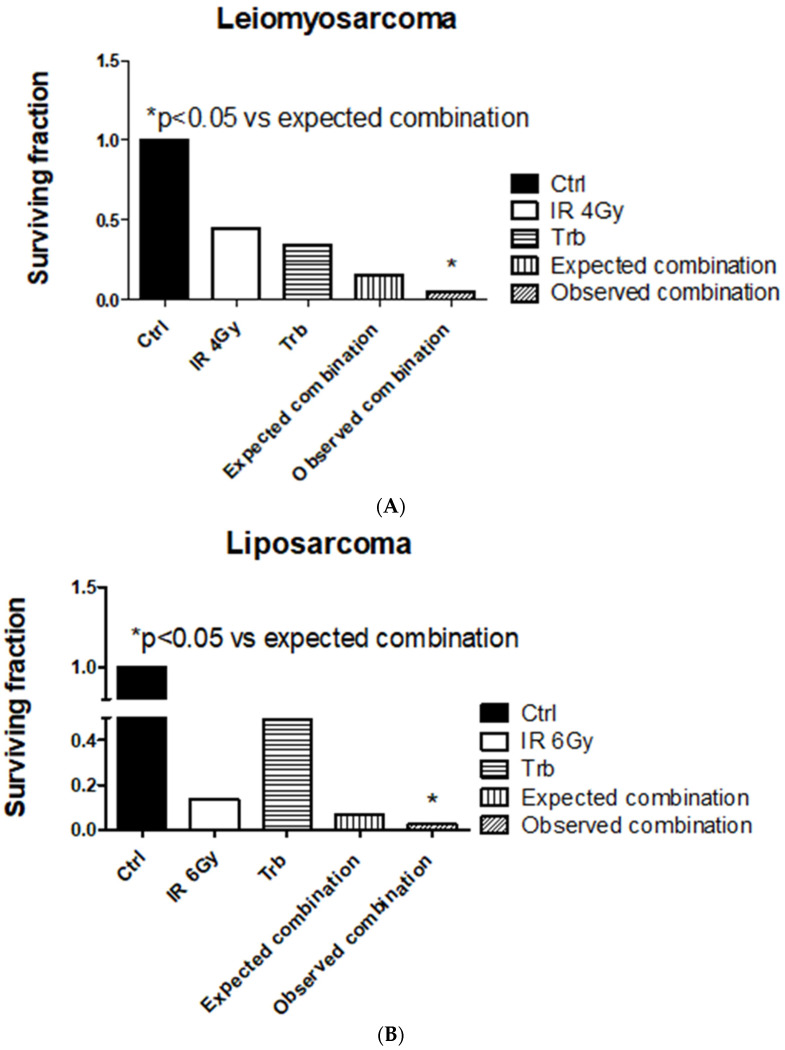
Trabectedin (trb) combined with radiation-induced synergistic effect in leiomyosarcoma (HS5.T) and liposarcoma (SW872) cells. (**A**) Leiomyosarcoma HS5.T. (**B**) Liposarcoma SW872. Observed combination: surviving fraction observed after irradiation (IR) with trabectedin (trb) treatment. Expected combination: surviving fraction calculated as the product of the surviving fractions observed after treatment with trabectedin or radiation alone. Data are shown as the means ± SEM. * *p* ≤ 0.05 vs. the expected surviving fraction. Ctrl: control.

**Figure 4 ijms-23-14305-f004:**
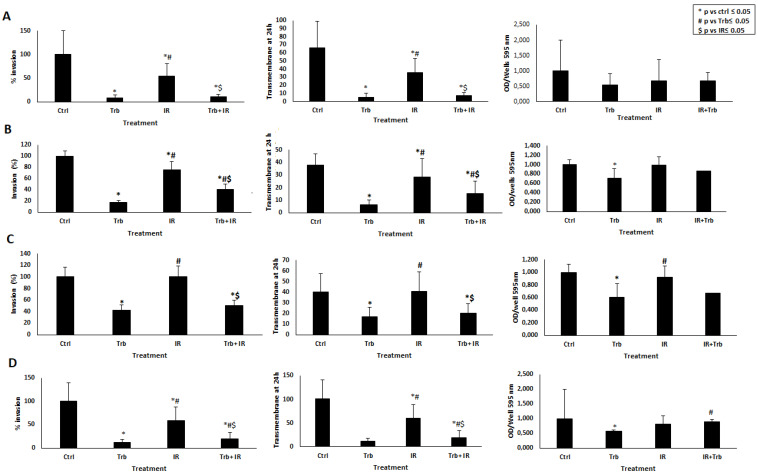
Invasion assay in STS cell lines. The matrigel cell invasion assay was performed in each STS cell line after irradiation (IR) and/or in the presence of trabectedin (trb). Columns: capability of invasion reported in % (left) and at 24 h (middle), quantification of 595 nm with a plate reader (right) for rhabdomyosarcoma RD (**A**), liposarcoma SW872 (**B**), leiomyosarcoma HS5.T (**C**), and fibrosarcoma HS 93.T (**D**) * *p* ≤ 0.05 vs. control, # *p* ≤ 0.05 vs. trb, $ *p* ≤ 0.05 vs. IR.

**Figure 5 ijms-23-14305-f005:**
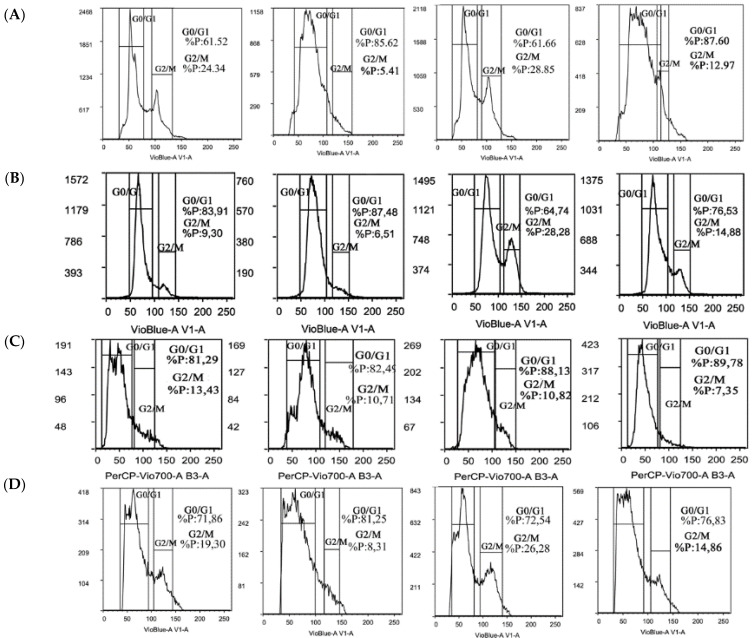
Cell cycle progression in response to trabectedin. Representative flow cytometry analysis showing cell cycle progression in each STS cell line after treatment with trabectedin and/or 4 Gy. (**A**–**D**) Rhabdomyosarcoma RD, liposarcoma SW872, leiomyosarcoma HS5.T, and fibrosarcoma HS 93.T cells, respectively. Analysis was performed by quantifying DAPI nuclear staining using flow cytometry. DNA content and the number of events were analyzed 24 h after treatment. Relative percentage in the G0/G1, G2/M, and S cell phases are plotted after FACS analysis. (**E**) Relative percentage in the G0/G1, G2/M, and S cell phases are plotted after FACS analysis (* *p* < 0.05 vs. control, # *p* < 0.05 vs. trabectedin, $ *p* < 0.05 vs. irradiation).

**Figure 6 ijms-23-14305-f006:**
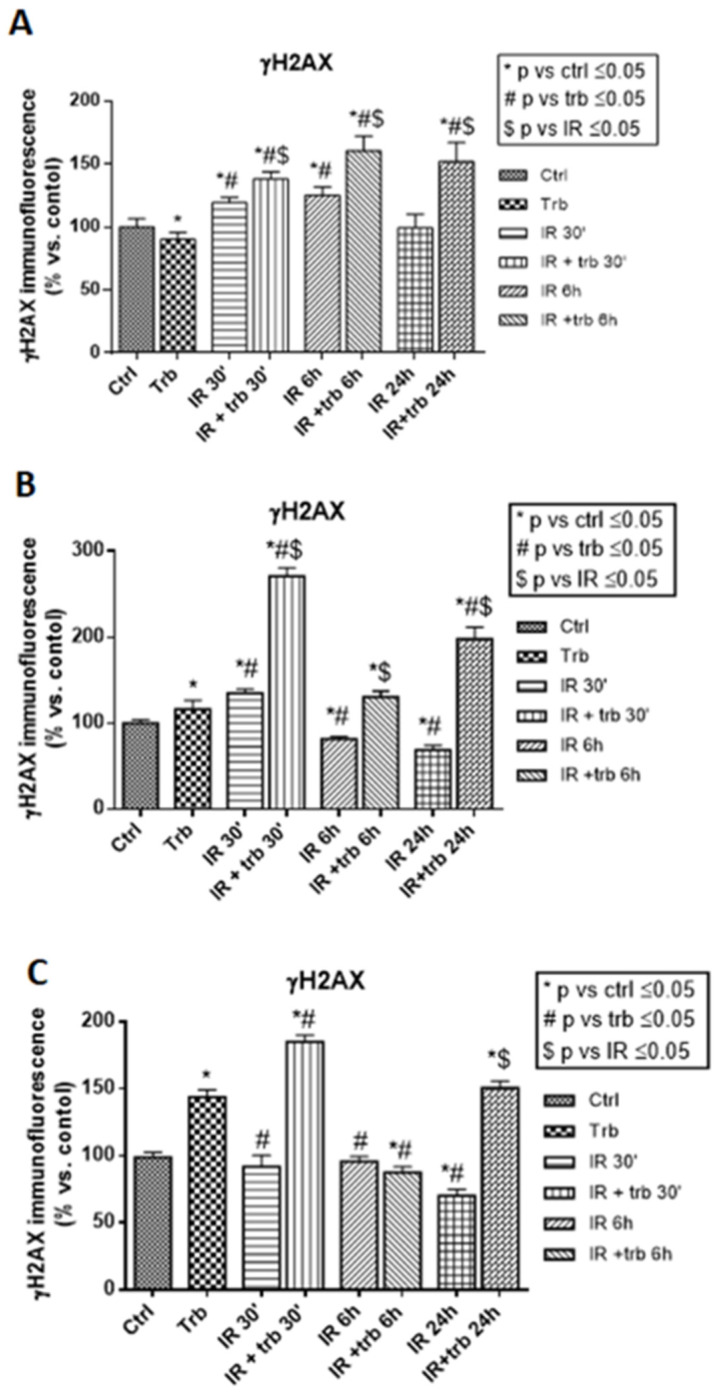
Trabectedin significantly increased the mean immunofluorescence emitted by γ-H2AX foci in irradiated rhabdomyosarcoma RD (**A**), liposarcoma SW872 (**B**), and leiomyosarcoma HS5.T (**C**) cell lines. The mean γ-H2AX immunofluorescence is reported as a percentage of the control and shown as the mean ± SEM. * *p* ≤ 0.05 vs. control (ctrl); # *p* ≤ 0.05 vs. trabectedin (trb); $ *p* ≤ 0.05 vs. irradiation (IR).

**Figure 7 ijms-23-14305-f007:**
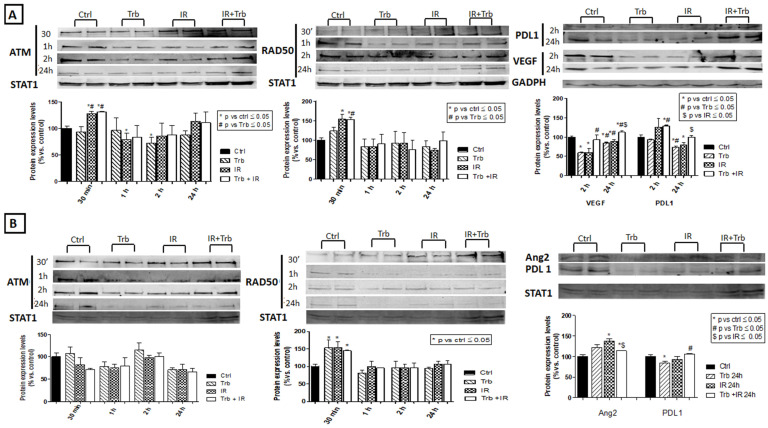
Trabectedin affects protein expression in leiomyosarcoma HS5.T (**A**) and liposarcoma SW872 (**B**). Western blot analysis of the expression of ATM, RAD50, VEGF, Ang-2, and PDL-1 in leiomyosarcoma (HS5.T) and liposarcoma SW872 cells treated with the vehicle (control, ctrl), 4 Gy irradiation (IR) alone, or with trabectedin (Trb), respectively. GAPDH and STAT1 were used as loading controls. Data reported as the percentage versus the respective control (set at 100) are shown as the mean ± SEM. * *p* ≤ 0.05 vs. control, # *p* ≤ 0.05 vs. trabectedin, $ *p* ≤ 0.05 vs.IR.

## Data Availability

The data that support the findings of this study are available from the corresponding author upon reasonable request.

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
