# Peer review of "Radiosensitizing Effect of Trabectedin on Human Soft Tissue Sarcoma Cells"

_ijms, 2022, doi:10.3390/ijms232214305_

Round 1

Reviewer 1 Report

The manuscript provided by Loi et al. (ijms-2005819) investigated the effect of trabectin on X-ray efficacy in four different subtypes of soft tissue sarcomas in an in vitro model. The authors attempted to investigate the mechanism responsible for the synergistic effect by examining the mechanisms of inter alia DNA damage repair, neoangiogenesis and immunity avoidance. Loi et al. demonstrated the sensitization of cells of two STS subtypes by TR to IR. This research is very detailed and innovative, the results of this work might be of interest to the readers of IJMS. However the reviewer has a few concerns that need to be supplemented and clarified.

1. Many Figures (especially Figure 5) are quite illegible due to the very small font of the captions. In addition, in the case of Figure 5, it would be valuable to add bar graphs with the standard deviation marked out showing what percentage of cells were at different phases of the cell cycle in each of the samples tested.

2. What was the method used to test TR cytotoxicity? It is not specified in the methodology. Was it MTT, MTS or some other method?

3. The Western Blot study also raises reservations. In Figure 7B, the bands corresponding to the STAT1 as reference protein are very faintly visible. Moreover, it is clearly visible that the STAT1 protein band in the "IR + TR" sample is thicker than the others, which means the whole protein loading amount is different. Therefore, I would suggest remeasuring it to make it more convincing.

4. How was the clonogenic assay performed exactly? What was the incubation time of the tested cell lines with the compounds? There is nothing in the reference given by the authors regarding this test.

5. No signature of figures in the supplements.

Kind regards,

Author Response

We would like to thank the reviewer for his/her comments that improve the quality of our study.

Please find enclosed the point-by-point response to the reviewer’s comments below.

We hope we fulfilled reviewer’s requests and that the revised manuscript will be accepted for publication.

1) Many Figures (especially Figure 5) are quite illegible due to the very small font of the captions

Re: We apologize with reviewer, we modified the quality of the figures by increasing the font of the captions. We also add a bar graph about the different percentage observed in the different phases of the cell, as suggested. Here below the graph.

 (see enclosed document)

We added the sentence “Relative percentage in G0/G1, G2/M and S cell phases were plotted after FACS analysis (*P< 0.05 vs. control, # P<0.05 vs. Trabectedin, § P< 0.05 vs. irradiation)

2) What was the method used to test TR cytotoxicity? It is not specified in the methodology. Was it MTT, MTS or some other method?

Re: The IC50 values were measured with an MTS assay. We add “An MTS assay was used to measure IC50” in the section of materials and methods.

3) The Western Blot study also raises reservations. In Figure 7B, the bands corresponding to the STAT1 as reference protein are very faintly visible. Moreover, it is clearly visible that the STAT1 protein band in the "IR + TR" sample is thicker than the others, which means the whole protein loading amount is different. Therefore, I would suggest remeasuring it to make it more convincing.

Re: For WB experiments were used two different loading controls, GADPH and STAT1, depending on the molecular weights of the protein to avoid overlapping signal. STAT1 reported in the figure 7B is only representative as the showed graphs are the results of the densiometric mean calculated for each different protein analyzed.  Here below to strength our results we reported another STAT-1 and also GADPH.

  (see enclosed document)

4) How was the clonogenic assay performed exactly? What was the incubation time of the tested cell lines with the compounds? There is nothing in the reference given by the authors regarding this test.

Re: A clonogenic assays lasts about 15 days depending on the cell lines. We apologize with the reviewer, we put the wrong number of the reference, we corrected with the right one, number [31]. Here below the reference we refer to:

Mangoni M, Sottili M, Gerini C, Desideri I, Bastida C, Pallotta S, Castiglione F, Bonomo P, Meattini I, Greto D, Cappelli S, Di Brina L, Loi M, Biti G, Livi L. A PPAR-gamma agonist protects from radiation-induced intestinal toxicity. United European Gastroenterol J. 2017 Mar;5(2):218-226. doi: 10.1177/2050640616640443. Epub 2016 Jul 8. PMID: 28344789; PMCID: PMC5349355

5) No signature of figures in the supplements.

Re: We apologize with the reviewer, we reported in the manuscript (below the conclusion section) the two figure legends of supplementary data, as shown below.

-Supplement Figure 1.  A representative image of three experiments of invasion assay is shown for each STS cell line, 40X original magnification, respectively rhabdomyosarcoma (A), liposarcoma (B), leiomyosarcoma (C) and fibrosarcoma (D). Control (ctrl); irradiation (IR); Trabectedin (Trb).

- Supplement Figure 2. Immunofluorescence staining of γ-H2AX after 4 Gy irradiation at different time points in rhabdomyosarcoma, liposarcoma and leiomyosarcoma respectively. Magnification: 40X. Control (ctrl), irradiation (IR), Trabectedin (Trb) and expected versus observed IR+Trb combination in leiomyosarcoma (A) and liposarcoma (B)

Reviewer 2 Report

In this manuscript, the authors studied the function of trabectedin on radio sensitivity with cell assays. The authors reviewed the current usage and knowledge on trabectedin in the treatment of STS, and explained the gap on the function of trabectedin combined with IR in cell radio sensitivity in in vitro models. To tackle this question, the authors first investigated the effects of trabectedin in cytotoxic effect of IR, and confirmed that the drug reduced invasiveness by the treatment of IR in STS cells and increased the H2AX foci while influenced the cell cycle progression. The authors further studied the expression of potentially related proteins including DNA damage response proteins, angiogenic factors and immune checkpoint proteins. Overall, the study is well designed and the results are very informative to the current studies on trabectedin and provide insights in potential function of the combined treatment of trabectedin and IR.

There are some minor suggestions before publication. The authors used four STS cell lines to confirm the findings and the effects of trabectedin and IR are slightly different in some of the cell lines. This may indicate the different function of trabectedin in different types of STS. If the authors can compare the effects of trabectedin in different STS, and other tumors in discussion, this will enlighten more in the trabectedin and IR treatments. In summary, the manuscript is suggested to be accepted. 

Author Response

Response to Reviewer #2's

Re: We would like to thank the reviewer for his/her comments that improved the quality of our study. We are already considering implementing our results to better understand how trabectedin, together with radiation, could induce a tumor-specific response. Then, it would be really interesting to extend these studies to other tumor types.  Especially due to the fact that, at the steady state, there not so many papers published focus on combined treatment of trabectedin and irradiation except in soft tissue sarcomas. As reported in the discussion, we found only one interesting paper aimed to examine the cytotoxic and radiosensitizing effects of trabectedin on two human epithelial tumor cell lines in vitro, and its effects on DNA repair capacity.

Round 2

Reviewer 1 Report

The authors of the manuscript made the required corrections and explained comments and questions. The reviewer accepts the manuscript in its current form.